# Accuracy of the SenseWear Armband during Short Bouts of Exercise

**DOI:** 10.3390/sports12040093

**Published:** 2024-03-26

**Authors:** Ryan D. Wedge, Mike McCammon, Stacey A. Meardon

**Affiliations:** 1Department of Physical Therapy, College of Allied Health Sciences, East Carolina University, Greenville, NC 27834, USA; 2Department of Kinesiology, College of Health and Human Performance, East Carolina University, Greenville, NC 27834, USA; mccammonm@ecu.edu

**Keywords:** gait, locomotion, metabolism, physiological monitoring

## Abstract

A goal of mobile monitoring is to approximate metabolic energy expenditure (EE) during activities of daily living and exercise. Many physical activity monitors are inaccurate with respect to estimated EE and differentiating between activities that occur over short intervals. The objective of our study was to assess the validity of the SenseWear Armband (SWA) compared to indirect calorimetry (IC) during short intervals of walking and running. Twenty young, fit participants walked (preferred speed) and ran (75%, 85%, and 95% of predicted VO_2max_ run speeds) on a treadmill. EE estimates from IC, SWA, and prediction equations that used the SWA, speed, and heart rate were examined during each 4 min interval and across the whole protocol (Total). The level of significance was *p* < 0.05. The SWA overestimated EE relative to IC by 1.62 kcal·min^−1^ while walking and 1.05 kcal·min^−1^ while running at 75%. However, it underestimated EE at the 85% (0.05 kcal·min^−1^) and 95% (0.92 kcal·min^−1^) speeds, but not significantly, and overestimated total EE by 28.29 kcal. Except for walking, our results suggest that the SWA displayed a good level of agreement (ICC = 0.76 to 0.84) with IC measures. Activity-specific algorithms using SWA, speed, and heart rate improved EE estimates, based on the standard error of the estimates, but perhaps not enough to justify extra sensors. The SWA may enable EE estimation of locomotion outside the laboratory, including those with short bouts of high intensity activity, but continued development of the SWA, or devices like it, is needed to enable accurate monitoring.

## 1. Introduction

Tracking energy expenditure (EE) accurately during exercise and training is important to ensure adequate energy availability to support task demands and basic physiological functions [1] and can serve as input to training load estimations [2]. A goal of mobile monitoring is to approximate metabolic EE during activities of daily living and exercise. Many of the currently used physical activity monitors are inaccurate with respect to estimated EE and differentiating between activities (e.g., cycling, stair climbing, walking at inclines) because of difficulties in separating intensity levels [3,4]. Further, accuracy is limited for sports characterized by high intensity and intermittent intervals [5]. EE estimations could be improved by combining sensor data from different physiological signals [6] with better prediction equations [7]. Small sensors combined with prediction algorithms may be the future of accurate EE estimations during intermittent, high intensity activity outside of the laboratory, approaching the accuracy of open circuit spirometry methods for indirect calorimetry (IC), and providing researchers and clinicians witha valuable tool for non-lab use.

Activity-specific EE is usually estimated with IC, which measures the rate of oxygen intake and carbon dioxide expiration. IC uses portable or static systems, requiring a mask or use of a bite valve and nose clip that is cumbersome and not conducive for long data collections. Accurate estimation is also delayed with IC, which requires at least 4 min of steady-state activity; the first 3 min are not used to calculate EE because gas exchange is normalizing to the activity intensity, and the final 1–2 min are used. During free-living activities, most bouts of walking consist of less than 4 min at one set pace [8]. Further, many field and court sports involve short bouts of locomotor activities ranging from walking to sprinting. The time delay needed for accurate EE using IC exceeds locomotor durations commonly associated with free-living and training activities. To move outside of the laboratory, sensors are needed that can estimate activity-specific and cumulative EE, with the accuracy of IC, but without the physiological delay to attain steady state.

Monitoring daily activities and training has become easier with new wearable technology and sensors. Devices are smaller, portable, and more comfortable for the user, allowing the physiological data to be collected without biasing the data due to an unnatural laboratory setting or laboratory equipment. Devices that rely on accelerometer data are less sensitive and accurate during sedentary and light intensity activity, relying on linear regression equations to approximate EE [3,4]. Other physiological signals, such as heart rate and skin temperature, may provide information about EE when the device itself is not moving (i.e., stationary bicycling). The SenseWear Armband (SWA), developed by BodyMedia, is a relatively unobtrusive sensor worn on the upper arm during physical activity and collects information about near-body ambient temperature, skin temperature through a proprietary heat flux sensor, triaxial acceleration, and galvanic skin response, which varies due to physical and emotional stimuli. Output from each of these signals is available with the SWA Professional software (Version 8.1.1.36), but how the software uses these measures along with demographic characteristics (i.e., age, weight, height, sex, handedness, smoker or non-smoker) in a proprietary algorithm to estimate EE in free living conditions is unknown. The SWA outputs EE estimations each minute using this unshared proprietary algorithm.

Previous studies have assessed the SWA during long bouts of activity but the validity of it during short bouts of physical activity from light to vigorous intensity is not fully known [9,10]. In the few studies that have examined the SWA across intermittent bouts of exercise consisting of varying intensities, device accuracy has been questioned [11,12,13,14]. The results from Gastin et al. [11] showed a 36.7% difference for walking, 14.9% difference for running (12 km·h^−1^), and 36% difference for circuit training when comparing the SWA to IC. Also, the results from van Hoye et al. [12] suggest that the accuracy of the SWA compared to IC worsens with increasing speeds of an incremental exercise test; the error increased from 2–5% at 1.5 m·s^−1^ to 39–46% at near max 4.0–4.5 m·s^−1^, with the largest errors (>19%) seen at running speeds ≥3.0 m·s^−1^ [12]. These results are consistent with other studies that report that the SWA underestimates EE during high-intensity physical activity [12,15,16,17]. One reason may be that its proprietary algorithm does not account for heart rate (HR), which has been shown to be an important predictor of EE using sensor data [6]. Because EE is directly related to HR and speed with running, incorporating additional sensors capable of measuring HR and speed could improve the accuracy of EE estimations [18,19] and represents a gap to be addressed. 

The purpose of the study was to assess the validity of the SWA relative to IC during short intervals of walking and running at various intensities (75, 85, and 95% predicted VO_2_max). Secondarily, the present study sought to examine the effects of SWA prediction equations with and without HR and speed data to IC estimations. It was hypothesized that the SWA would demonstrate moderate to good agreement with IC across various locomotor intensities. It was also hypothesized that prediction equations incorporating heart rate and speed with the SWA data would increase the accuracy of EE estimates compared to SWA alone.

## 2. Materials and Methods

### 2.1. Participants

A convenience sample for this laboratory study was recruited to examine the differences between metabolic energy expenditure (EE) estimates of the SenseWear Armband (SWA) and indirect calorimetry (IC) at a preferred walking speed and standardized running speeds. The study was conducted in accordance with the Declaration of Helsinki, and all procedures performed during the study were approved by the local University Institutional Review Board (UMCIRB 15-002097) and all participants provided written informed consent after being told the risks and benefits of the study. Twenty physically active adults, free of any known disorders (e.g., cardiovascular, neuromuscular) that affected their ability to exercise or were exacerbated by exercise, volunteered for this study. To ensure safety with the study protocol, all volunteers were required to be recreationally active, defined as having a Perceived Functional Ability (PFA) score of ≥16/26 (i.e., 11–12 min per mile running pace), a Physical Activity Rating (PAR) of ≥6/10 (i.e., runs ≥ 5 miles per week), and a treadmill comfort score of ≥7/10 (i.e., completely comfortable). Prior to participation, participants were screened for all inclusion criteria and were interviewed using the AHA/ACSM Health/Fitness Facility Pre-participation Screening Questionnaire to ensure low risk classification according to the AHA/ACSM Risk Stratification guidelines.

### 2.2. Procedures

To standardize effort during treadmill running conditions, participants’ VO_2_max values were predicted using the George Non-Exercise Equation (R = 0.86 and SEE = 3.34) [20] (Equation (1)):(1)$$\begin{array}{c} VO_{2}max_{pred}=45.513+\left(6.564\times Sex\right)-\left(0.749\times BMI\right)+\left(0.724\times PFA\right)\\ \;\;\;\;\;\;+\left(0.788\times PAR\right)\; \end{array}$$
where sex = 1 for males, 0 for females. Running speeds corresponding to 75, 85, and 95% of participants predicted VO_2_ max were then estimated using a metabolic equivalent running equation by solving for speed, with speed units corresponding to m·min^−1^ [21] (Equation (2)): (2)$$VO_{2}=\left(0.2\times speed\right)+\left(0.9\times speed\times grade\right)+3.5$$
where grade was 0%. A traditional VO_2_max test was not used because the purpose of the study was to validate the SWA at different levels of exertion, versus examining the effects of VO_2_max, and maximal capacity testing of participants would have been an unnecessary burden. 

After successful clearance for exercise testing and determination of treadmill running speeds, the SWA Mini-Fly (Model MF-SW, BodyMedia, Pittsburgh, PA, USA) was configured, placed on the triceps of the left arm, and initialized prior to testing using demographic data provided by each participant, per manufacturer instructions. For IC, participants were fitted with a facemask that was connected to the metabolic cart (ParvoMedics TrueOne 2400, Sandy, UT, USA) to measure oxygen intake and carbon dioxide output levels through open circuit spirometry. Data from the SWA and IC were continuously recorded as the participants completed 5 conditions at 0% incline: walking and running at a self-selected pace and at running speeds corresponding to 75, 85, and 95% VO_2_max_pred_. Each condition lasted for 4 min, followed by an active recovery period of at least 3 min in between each condition. During recovery, participants walked at their pre-determined preferred walking speed. The order of running conditions was randomized to minimize order effects. During the testing protocol, SWA and IC data were collected continuously with timestamps based on GMT-5 to allow for time synchronization between devices. HR was recorded at each minute from a Polar™ heart rate monitor (Polar RS800CX; Polar Electro Ov, Kempele, Finland). 

### 2.3. Dependent Variables

Time stamped SWA, IC, and HR data were exported for analysis. The SWA Professional software (Version 8.1.1.36) provided minute-by-minute data from each sensor (triaxial accelerations, galvanic skin response, near body temperature, and skin temperature) as well as EE (kcal·min^−1^) determined using a proprietary algorithm. The ParvoMedics software (OSUW 4.3) provided oxygen consumption per kilogram every 20 s, which was used to calculate METS·min^−1^ and kcal·min^−1^ (Equations (3) and (4)).
(3)$$METS\cdot min^{-1}=\left(VO_{2}\cdot \;kg^{-1}\right)/3.5$$
(4)$$kCal\cdot min^{-1}=\left(\left(METS\times 3.5\times mass\right)/1000\right)\times 5\;$$Using time stamps, each 4 min interval was identified and corresponding IC, SWA, and HR data were extracted and averaged for statistical analysis. 

### 2.4. Statistical Analysis

Analysis of variance (ANOVA) models were used to assess differences in mean EE between IC and the SWA for each condition as well as the total exercise session (α ≤ 0.05) with Holm–Bonferroni corrections for multiple comparisons [22]. Magnitude of differences were examined using Cohen’s d_z_ effect sizes (es) [23] with values less than 0.5, between 0.5 and 0.8, between 0.8 and 1.2, between 1.2 and 2.0, and greater than 2.0 to be indicative of small, medium, large, very large, and huge effects, respectively [24]. Additionally, for each condition, intraclass correlations (ICC (3,1)) were computed to examine the level of absolute agreement between the technologies. ICC values less than 0.5, between 0.5 and 0.75, between 0.75 and 0.9, and greater than 0.90 were considered to be indicative of poor, moderate, good, and excellent levels of agreement, respectively [25]. 

Linear regression analysis using the enter method was used to develop EE prediction equations for each speed condition based on (1) SWA alone, (2) SWA and average HR, (3) SWA and treadmill speed, and (4) SWA, HR, and treadmill speed. A second-order Akaike information criterion(AIC_c_), corrected for small sample size, was used to identify the models that explained the greatest amount of variation in EE obtained from IC with the fewest variables [26]. Adjusted coefficient of multiple determination (Adj R^2^) and standard errors of the estimate (SEE) were secondarily examined to evaluate goodness-of-fit [26]. Statistical analyses were performed in SPSS version 25 (IBM SPSS Statistics v25, IBM Analytics, Armonk, NY, USA).

## 3. Results

Twenty participants, ten males (24.0 ± 2.9 [years], 177.3 ± 4.9 [cm], 80.5 ± 12.5 [kg], 52.8 ± 4.4 VO_2_max_pred_ [l·kg^−1^·min^−1^], 0.98 ± 0.3 preferred walking speed [m·s^−1^]) and ten females (22.3 ± 2.7 [years], 165.3 ± 6.2 [cm], 61.7 ± 0.4 [kg], 47.8 ± 4.7 VO_2_max_pred_ [l·kg^−1^·min^−1^], 1.35 ± 0.3 preferred walking speed [m·s^−1^]) participated in this study. 

Condition specific outcomes are displayed in Figure 1, with statistics reported in Table 1 and described below. EE values between devices differed only during walking and low speed running and demonstrated moderately good to good absolute agreement (ICC = 0.53–0.79).

Table 2 outlines the outcomes from the linear regression analysis using EE from the SWA alone and the SWA with average HR, treadmill speed, and both average HR and treadmill speed to predict EE obtained from IC. The results by condition are described below. Scatter plots with lines of best fit of linear regression models suggest that all models performed comparably well apart from walking (Figure 2). Prediction equations for all models can be found in Table 3. 

Walking Outcomes: During 4 min walking intervals, the SWA overestimated average EE relative to IC by 1.62 kcal·min^−1^, 95% CI[1.14, 2.10], with moderate absolute agreement observed (Figure 1, Table 1). The linear regression results indicate that the model with the SWA alone explained the greatest amount of variation in IC EE using the fewest possible predictors (i.e., lowest AIC_c_ values). The addition of HR and/or speed did not improve model outcomes beyond that of the SWA walking-specific prediction (Table 2).

75% VO_2_max_pred_ speed Running Outcomes: The SWA also overestimated EE during running at 75% VO_2_max_pred_ speed by 1.05 kcal·min^−1^, 95% CI[0.26, 1.98], but absolute agreement was good (Figure 1, Table 1). Linear regression models with SWA and speed as predictors maximized the variance accounted for (Adj. R^2^) and minimally improved the accuracy of the estimates (SEE). However, the lowest AIC_c_ values were observed with the linear regression model that included only the SWA to predict IC EE (Table 2).

85% VO_2_max_pred_ speed Running Outcomes: Device differences were not as apparent during the intervals when participants ran at 85% (0.05 kcal·min^−1^, 95% CI[−1.19, 1.08]). Absolute agreement at 85% VO_2_max_pred_ running speed was good (Figure 1, Table 1). Like lower speed running, models with SWA and speed as predictors maximized the variance accounted for and minimized SEE. Also, like lower speed running, the lowest AIC_c_ values were observed with the linear regression model that included the SWA alone to predict IC EE (Table 2).

95% VO_2_max_pred_ speed Running Outcomes: Device differences were also less apparent during the intervals when participants ran 95% VO_2_max_pred_ running speeds (0.86 kcal·min^−1^; 95% CI[−1.78, 0.07]). Similar to other conditions, absolute agreement at 95% VO_2_max_pred_ running speed was good (Figure 1, Table 1). Like other running conditions, models with SWA and speed as predictors displayed higher Adj. R^2^ and lower SEE. However, unlike other walking and running conditions, linear regression AIC_c_ values for the 95% predicted VO_2_max_pred_ running speed were lowest when using SWA, HR, and speed as predictors for IC EE (Table 2).

Entire Protocol Outcomes: Across the entire protocol, the SWA tended to overestimate total EE by 28.48 kcal (95% CI[12.359, 44.59]) but the level of agreement was good (Figure 1, Table 1). The AIC_c_ was lowest when using SWA and HR as predictors (Table 2).

## 4. Discussion

The purpose of our study was to assess the concurrent validity of the SenseWear Armband (SWA) energy expenditure (EE) estimates relative to EE from indirect calorimetry (IC) during short bouts of walking and running. The study also sought to determine if prediction equations with SWA, HR, and speed data would improve the accuracy of EE estimates. Somewhat consistent with our first hypothesis, the SWA estimated EE for different exercise intensities with good accuracy across subjects, except walking. Additionally, including speed and HR with SWA data did not consistently improve EE predictions compared to SWA alone as expected. Except for the highest speed of running, the SWA alone explained the greatest amount of variation in EE derived from IC using the fewest possible variables. Including speed and HR in EE prediction equations along with SWA did explain more variation in EE, especially at higher speeds, but only slightly and not significantly. The inclusion of speed and HR in future devices depends on cost, unification into one device, and burden to the user.

The results from the few studies that have examined the SWA during short bouts of activity are conflicting [12,13]. In one study of a shuttle run test averaging ~7 min, the SWA demonstrated small but not statistically significant differences in EE (kcal·min^−1^) relative to portable IC with good correlations reported, despite differences seen during 1-to-2 min intervals [13]. Our results generally agree with these findings. However, in a study of an incremental treadmill exercise test with 3-to-5 min intervals, the levels of agreement between the SWA and IC were poor and the SWA tended to underestimate EE, especially as exercise intensity increased [12]. The inconsistent results across studies may be explained by differing test conditions, SWA models, and SWA software versions.

Our results also agreed with previous reports that the SWA overestimated EE at lower intensities [11,27,28] and underestimated EE at higher intensities [11,15,16,17]. The SWA EE estimates obtained using the SWA model and software in this study were not significantly different from IC during short bouts of high-speed running, suggesting utility for sports and training characterized by short bouts of linear, high-intensity activity. The speed-dependent positive and negative bias associated with SWA may offset each other over time, minimizing cumulative differences relative to IC when compared to small intervals. The study found the greatest absolute agreement between the SWA and IC in total EE (kcal) over the complete study protocol, despite overestimating total EE by 10%. This suggests that the SWA performed reasonably well during intermittent exercise with varying intensities. These results differ from Gastin et al., who reported up to 37% differences during a 15 min interval comprised of a 5 min sport circuit of intermittent linear and non-linear walking, running, and sprinting and seated recovery [11]. It is possible that the SWA has worse accuracy with non-linear activities. It may also struggle with post-exercise recovery periods. In a recent study of male rugby players using an earlier software version, the SWA EE estimates were only moderately correlated to IC during an intermittent exercise test and recovery period, with greater bias reported during recovery [14]. Further testing is needed with updated software to identify the need for algorithm refinement.

Activity-specific prediction equations may improve the accuracy of the SWA. In this study, the SWA alone explained the greatest amount of variation using the fewest possible predictors in all conditions except for the 95% run condition. However, based on the adjusted R^2^ and SEE values of linear regression prediction equations, the SWA estimates improved when using activity-specific predictive equations that included HR and/or speed [29]. This is similar to previous findings of activities lasting 20 to 30 min [9,30]. If the goal is to accurately estimate EE, such as with athletes or serious exercisers, extra sensors that capture data related to EE, like HR and speed [18,19], may be necessary. Ideally extra sensors would be integrated into one sensor to increase acceptance across users. Furthermore, the accuracy of EE estimates from devices with multiple physiological signals could be augmented using activity-specific equations or implementing machine learning approaches [7]. However, activity-specific algorithms are subject to accurate sensor detection of new activities, which is challenging [31,32], or manual user entry of activities, which increases user burden. Notwithstanding, the results from this study, combined with previous reports [30], suggest that an unobtrusive device capable of capturing multiple signals to estimate EE, like the SWA, may be responsive to the energetic demands of training characterized by intermittent high-intensity locomotor activities when compared to the well-known delays of expired gas exchange [15,22].

Our study was limited to a small sample size of young, relatively fit individuals, and our data can only be generalized to linear walking and running. We determined that 20 was a large enough sample to detect device differences based on an a priori power analysis and preliminary lab data but acknowledge the limitations of this sample size for ICC analysis and exploratory prediction equations. Studying a homogenous population limits generalizability. Additionally, sex-specific differences in SWA EE estimation have been reported [12]. Sex was accounted for in the estimated VO_2_max equations used for running speed determination and may be included the SWA proprietary algorithms. But due to sample size, sex differences in EE estimation were not examined. Further, because a VO_2_ max test was not performed, the results are limited to the relative versus absolute effects of exercise intensity on EE estimates from the SWA. Without a max test, the Critical Speed [33] and role of the slow component [34] could not be accurately assessed, which may have led to an underestimation of EE with IC. The speeds of each task in this study were based on known treadmill speed; thus, if speed is important for accurately estimating EE, the ability of SWA to sense speed and step counts needs to improve [29,35]. It is also possible that EE estimation may improve with integration of other physiological signals that were not collected (e.g., muscle activity from EMG, respiration rate). Lastly, because accelerometer location affects the measurement of movement characteristics [36], EE estimation may have been affected by the sensor location (left triceps) prescribed by the user manual.

Future studies will need to compare heterogenous populations, including sex effects, across different tasks to determine if the SWA can accurately estimate EE across a general population during activities involving short bouts of free-living conditions and exercise. Further, while the SWA may be a promising solution to estimating EE in the real-world for researchers and clinicians, to our knowledge it is no longer commercially available. BodyMedia was purchased by Jawbone in 2013 and the product was not pursued further; therefore, development of a new device incorporating multiple physiological signals may be needed.

Our results suggest that the SWA cannot replace laboratory measurements of IC but may provide reasonably accurate estimates of energy expenditure (EE) during short bouts of exercise, especially when combined with activity-specific algorithms and other physiological signals. Nonetheless, further development is needed. Compared to IC, wearable devices such as the SWA offer advantages like lower cost, reduced participant burden, and the ability to measure EE during various real-world activities and exercises, with less physiological delay. However, the SWA’s precise usefulness may depend on the setting. In our study, the mean difference between the SWA and IC was typically ~1 kcal·min^−1^ or less and within 10% of IC, except for short walking. Locomotion was the only activity examined, but the SWA and other commercially available sensors allow for EE estimation across different activities outside of the laboratory [31,37,38]. The capability to track EE in a reliable, ecological manner across multiple activities would allow researchers and clinicians to assess the effects of training on metabolic efficiency and energy availability during both short and long bouts of daily activities and exercise, as well as the effects of injury or disease on EE [2,39,40,41]. Valid wearable sensors with low participant burden could also help researchers and clinicians more readily identify EE effects of movement patterns across varying tasks, gait modification programs, and device settings (e.g., exoskeletons).

## 5. Conclusions

This study provides evidence that the SWA can estimate EE during short bouts of walking and running across various intensities in young, fit individuals. Except for walking, small absolute differences relative to IC were observed. The SWA has the potential to be a useful tool for monitoring workload and fitness progress outside the lab in sports and activities that involve short intervals, particularly within session, but further testing should be performed to determine repeated testing capabilities. Moreover, the SWA performed well at 85% and 95% of predicted VO_2_max, making it useful for higher intensity conditioning. The use of prediction equations that incorporate speed and heart rate may further enhance its accuracy, but further evaluation is needed.

## Figures and Tables

**Figure 1 sports-12-00093-f001:**
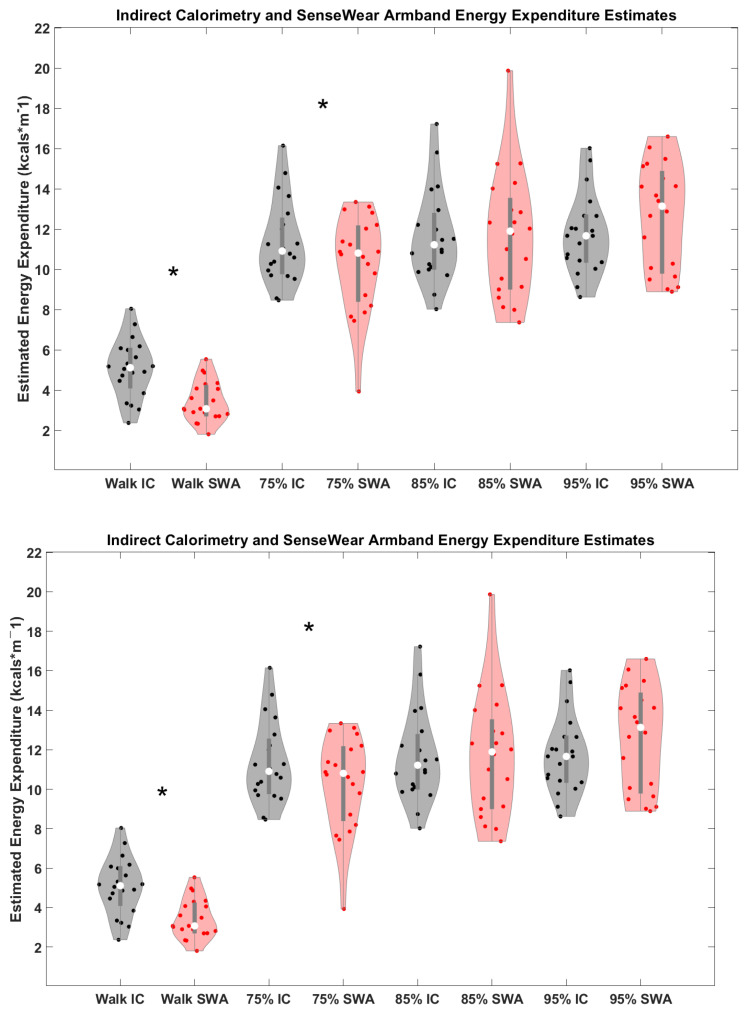
Indirect calorimetry and SenseWear Armband energy expenditure estimates: Violin plot of energy expenditure (EE) from indirect calorimetry (IC, black) and the SenseWear Armband (SWA, red) sensor across the preferred (pref) walk, and 75%, 85%, and 95% VO_2_max_pred_ running conditions showing the spread of the data around the median (open white circle) and end of first and third quartiles with the gray line in the center. The width of the plot shows the frequency of data around the estimated EE values. The SWA EE estimations for the preferred walk and 75% run conditions were significantly different from IC (*).

**Figure 2 sports-12-00093-f002:**
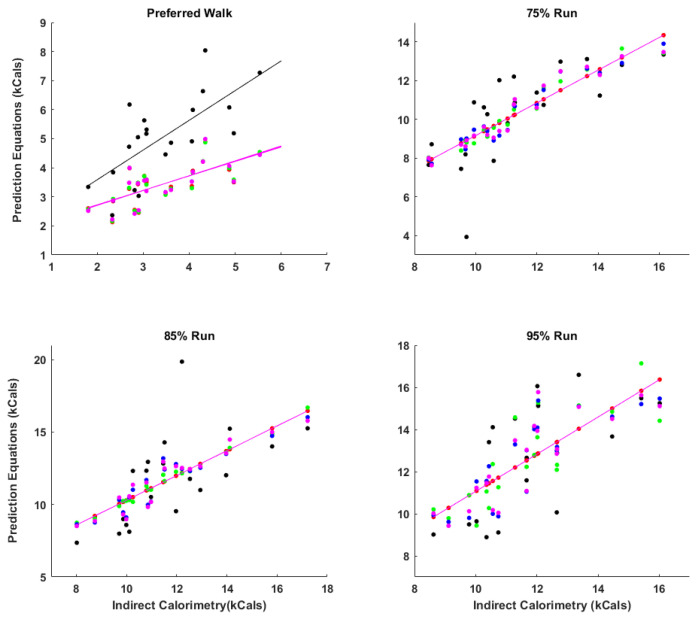
Regression lines for each prediction equation across four conditions. A linear relationship was observed for indirect calorimetry (IC) and the SenseWear Armband (SWA) sensor (black dots) as well as the 4 prediction equations examined: SWA (red dots), SWA plus heart rate (green dots), SWA plus speed (blue dots), SWA plus heart rate and speed (magenta ) to IC. The lines plotted are the linear regressions. Except for walking, significant overlap in least squared linear regression lines with the SWA sensor was observed. Regression equations for the figure can be found in Table 2.

**Table 1 sports-12-00093-t001:** Indirect calorimetry and SenseWear Armband energy expenditure estimates, mean ± 1 SD.

	IC (kcal)	SWA (kcal)	Mean Difference	Effect Size	*p*-Value	ICC [95% CI]
Pref Walk	3.44 ± 1.00	5.06 ± 1.44	1.62 ± 1.02	1.59	<0.01	0.53 [−0.24, 0.84]
75% Run	10.30 ± 2.40	11.35 ± 2.06	1.05 ± 1.68	0.63	0.01	0.79 [0.40, 0.92]
85% Run	11.71 ± 3.11	11.65 ± 2.30	0.05 ± 2.42	0.02	0.92	0.76 [0.39, 0.91]
95% Run	12.60 ± 2.61	11.74 ± 1.97	0.86 ± 1.97	0.43	0.07	0.76 [0.39, 0.90]
Total	284.95 ± 56.27	313.43 ± 54.46	28.48 ± 34.43	0.83	<0.01	0.84 [0.35, 0.95]

SWA = SenseWear Armband; HR = heart rate; EE = energy expenditure; Pref = preferred; IC = indirect calorimetry; CI = confidence intervals.

**Table 2 sports-12-00093-t002:** Results from linear regression analyses for the four models.

Models	Statistical Test	Pref Walk	75% Run	85% Run	95% Run	Total EE
SWA	R^2^	0.50	0.53	0.40	0.44	0.65
	Adj R^2^	0.47	0.50	0.37	0.41	0.63
	SEE	0.72	1.70	2.47	2.01	34.14
	AIC_c_	−10.30	28.63	39.99	36.85	144.96
SWA + HR	R^2^	0.50	0.56	0.41	0.53	0.68
	Adj R^2^	0.44	0.50	0.34	0.47	0.64
	SEE	0.74	1.69	2.53	1.90	33.87
	AIC_c_	−7.61	29.92	42.79	37.30	144.03
SWA + speed	R^2^	0.51	0.56	0.46	0.63	0.69
	Adj R^2^	0.45	0.51	0.39	0.58	0.66
	SEE	0.74	1.68	2.42	1.69	32.93
	AIC_c_	−8.07	29.31	41.62	32.61	144.15
SWA + HR + speed	R^2^	0.51	0.57	0.47	0.63	0.70
	Adj R^2^	0.42	0.49	0.37	0.57	0.64
	SEE	0.76	1.71	2.47	1.72	33.72
	AIC_c_	−4.10	32.13	42.72	27.98	145.78

SWA = SenseWear Armband; HR = heart rate; EE = energy expenditure; Pref = preferred.

**Table 3 sports-12-00093-t003:** Regression equations for indirect calorimetry (IC) derived EE from linear regression analysis for preferred walk, 75% run, 85% run, 95% run conditions and total energy expenditure (EE) across conditions. ICC(3,1) values with 95% confidence intervals (CI) provided to characterize the absolute agreement between predicted EE and IC values. ^†^ indicates the models with lowest AIC values.

Preferred Walk	Regression Equation	ICC [95%CI]
SWA ^†^	EE = 0.980 + 0.486 × SWA	0.81 [0.50, 0.92]
SWA + HR	EE = 0.702 + 0.471 × SWA + 0.004 × HR	0.81 [0.51, 0.93]
SWA + SP	EE = 0.406 + 0.473 × SWA + 0.497 × SP	0.82 [0.53, 0.93]
SWA + HR + SP	EE = 0.394 + 0.472 × SWA + 0.001 × HR + 0.490 × SP	0.82 [0.53, 0.93]
75% Run		
SWA ^†^	EE = 0.844 + 0.189 × SWA	0.82 [0.55, 0.93]
SWA + HR	EE = 6.183 + 0.782 × SWA − 0.031 × HR	0.84 [0.59, 0.94]
SWA + SP	EE = −2.725 + 0.796 × SWA + 1.041 × SP	0.84 [0.60, 0.94]
SWA + HR + SP	EE = 1.475 + 0.772 × SWA − 0.019 × HR + 1.021 × SP	0.85 [0.61, 0.94]
85% Run		
SWA ^†^	EE = 1.736 + 0.856 × SWA	0.75 [0.33, 0.89]
SWA + HR	EE = 4.126 + 0.846 × SWA − 0.014 × HR	0.74 [0.33, 0.90]
SWA + SP	EE = −4.138 + 0.812 × SWA + 1.953 × SP	0.78 [0.43, 0.91]
SWA + HR + SP	EE = 2.037 + 0.799 × SWA − 0.028 × HR + 1.595 × SP	0.78 [0.43, 0.91]
95% Run		
SWA	EE = 2.241 + 0.882 × SWA	0.77 [0.40, 0.91]
SWA + HR	EE = 9.810 + 0.828 × SWA − 0.041 × HR	0.89 [0.71, 0.96]
SWA + SP	EE = −6.691 + 0.793 × SWA + 2.706 × SP	0.88 [0.68, 0.95]
SWA + HR + SP ^†^	EE = −2.813 + 0.784 × SWA − 0.015 × HR + 2.369 × SP	0.88 [0.69, 0.95]
Total		
SWA	EE = 23.608 + 0.834 × SWA	0.90 [0.74, 0.96]
SWA + HR ^†^	EE = 135.185 + 0.786 × SWA − 0.680 × HR	0.90 [0.74, 0.96]
SWA + SP	EE = −81.181 + 0.774 × SWA + 44.484 × SP	0.91 [0.76, 0.96]
SWA + HR + SP	EE = −10.143 + 0.763 × SWA − 0.320 × HR + 36.656 × SP	0.91 [0.76, 0.96]

EE = energy expenditure; SWA = SenseWear Armband; HR = heart rate; SP = speed.

## Data Availability

The data necessary to reproduce the results are available on figshare with the DOI: https://doi.org/10.6084/m9.figshare.24199527.

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
