# Peer review of "Accuracy of the SenseWear Armband during Short Bouts of Exercise"

_sports, 2024, doi:10.3390/sports12040093_

Round 1

Reviewer 1 Report

Comments and Suggestions for Authors

This study evaluates the SenseWear Armband with respect to assessing energy expenditure during physical activity.  It is valuable and necessary to test such devices and assess their utility and accuracy and this provides substantial imperative for this study.  Although the study design was good and the results novel and intriguing certain criticisms temper enthusiasm for this work, at least as currently presented.

Principal among these criticisms are:

1.      It would have been good to measure VO2max and likely Critical Speed to better evaluate and set the exercising running intensities.  In part this would enable evaluation of the energetics contribution of the VO2 slow component which can account for as much as 5-8 kcal/min of exercise – which is substantial. This needs to be addressed for the reader. Use of %VO2max implies that a steady state VO2 can be achieved at the speeds.

Jones AM et al. Critical power: implications for determination of V˙O2max and exercise tolerance. Med Sci Sports Exerc. 2010 Oct;42(10):1876-90.

2.      Throughout (e.g. Lines 25, 189,) the writing and conclusions are overly judgemental.  Please rewrite to allow the reader to asses whether agreement is “good.”

3.      Should equation #3 be rewritten to include RER and thus account for the actual energy equivalent of the substrate?

4.      Figure 1.  Could the dots be joined within subjects to demonstrate how many followed the overall pattern?

5.      Lines 286/7         This is a pretty damning statement and needs placing in context.

6.      Lines 302-           This statement reduces the value of this work.  Are there next generations made by other companies now on the market?

Minor

Lines

96           “were recruited” is redundant.  Delete.

122        “effects of VO2max” This would have been a strength and, in the reviewer’s opinion, worth the subjects’ and investigators’ time and effort.

201        Pick either “minimized” or “improved” but not both.

328-       Lack of consideration of slow component behavior temper the strength of these conclusions.  Please reword in line with consideration of the literature in this regard.

Author Response

Reviewer 1

This study evaluates the SenseWear Armband with respect to assessing energy expenditure during physical activity.  It is valuable and necessary to test such devices and assess their utility and accuracy and this provides substantial imperative for this study.  Although the study design was good and the results novel and intriguing certain criticisms temper enthusiasm for this work, at least as currently presented.

Thank you for reviewing our manuscript and providing feedback.

Principal among these criticisms are:

  1. It would have been good to measure VO2max and likely Critical Speed to better evaluate and set the exercising running intensities.  In part this would enable evaluation of the energetics contribution of the VO2 slow component which can account for as much as 5-8 kcal/min of exercise – which is substantial. This needs to be addressed for the reader. Use of %VO2max implies that a steady state VO2 can be achieved at the speeds.

Jones AM et al. Critical power: implications for determination of V˙O2max and exercise tolerance. Med Sci Sports Exerc. 2010 Oct;42(10):1876-90.

Thank you for the comment. We did not perform a max test because the purpose of the study was to compare the two measurement techniques at the same time, and we used standardized procedures to determine the predicted VO2max. The hope of using the SenseWear Armband is that it could accurately estimate energy expenditure at any intensity, whether it is a predicted 75% VO2max pace or an actual 75% VO2max pace from a max test. We used standardized techniques of approximating energy expenditure with the Brockway (1987) equation and in non-laboratory situations, we would not be standardizing speeds in the field to get free living locomotion. The runners stabilized their VO2 output as evident by a visible plateau while collecting and VO2 not having a greater variability than 2.0 ml/kg/min.

We do acknowledge a limitation of not doing a max test and then accurately finding a Critical Speed and assessing the VO2 slow component. We have changed the limitations section to address this for readers. It now states:

“Further, because a VO2 max test was not performed, the results are limited to the relative versus absolute effects of exercise intensity on EE estimates from the SWA. Without a max test, the Critical Speed[33] and role of the slow component[34] could not be accurately assessed, which may have led to an underestimation of EE with IC.”

  1. Throughout (e.g. Lines 25, 189,) the writing and conclusions are overly judgemental.  Please rewrite to allow the reader to asses whether agreement is “good.”

Thank you for the comment. When we used words such as “good”, we were doing that based on the ICC results. We have made substantial changes to the manuscript based on other reviewer’s comments and hopefully the use of words such as “good” are more closely linked to the statistical results, such as in the abstract:

“Except for walking, our results suggest that the SWA displayed a good level of agreement (ICC=0.76 to 0.84) with IC measures.”

  1. Should equation #3 be rewritten to include RER and thus account for the actual energy equivalent of the substrate?

The equation used is based on Jetté et al. (1990) to calculate METs when VO2 is being measured, and then converted that to kcals.

Jetté M, Sidney K, Blümchen G. Metabolic equivalents (METS) in exercise testing, exercise prescription, and evaluation of functional capacity. Clin Cardiol. 1990 Aug;13(8):555-65. doi: 10.1002/clc.4960130809. PMID: 2204507.

  1. Figure 1.  Could the dots be joined within subjects to demonstrate how many followed the overall pattern?

Thank you for the comment but we think that would remove the clarity between IC and SWA across conditions with the violin plot, including how it could obscure the median and quartiles.

  1. Lines 286/7         This is a pretty damning statement and needs placing in context.

Thank you for the comment. That statement was based on previous reviews in which we discussed the SWA’s ability to detect group (e.g., sex) effects. Based on the extensive edits made to this revision, we think the passage is out of context and has been removed.

Moreover, while the accuracy of SWA may not be high enough to assess between subject differences in this study’s sample of participants, it may be sufficient for repeated testing given the high reliability reported in the literature [27,32].

  1. Lines 302-           This statement reduces the value of this work.  Are there next generations made by other companies now on the market?

Based on an online search, it looks like Jawbone did not pursue another iteration when they bought this technology. Someone could try to combine these sensors with their own algorithm for energy expenditure predictions. Using these signals in a unified sensor is enticing because it is less obstructive than indirect calorimetry or using things like electromyography to estimate energy expenditure.

Minor

Lines

 96           “were recruited” is redundant.  Delete.

Thank you for the comment, that section has been changed to now say:

“Twenty physically active adults, free of any known disorders (e.g., cardiovascular, neuromuscular) that affected their ability to exercise or were exacerbated…”

122        “effects of VO2max” This would have been a strength and, in the reviewer’s opinion, worth the subjects’ and investigators’ time and effort.

We understand the rationale for this comment but because of the study’s goal, we did not perform a VO2max test and are unable to recollect or do a VO2max test with these participants.

201        Pick either “minimized” or “improved” but not both.

Thank you for the comment. The results have gone through a significant revision based on another reviewer’s comments. Here is a passage from the revision that addresses this issue.

75% VO2maxpred speed Running Outcomes. The SWA also overestimated EE during running at 75% VO2maxpred speed by 1.05 kcal·min-1, 95% CI[0.26, 1.98], but absolute agreement was good (Fig. 1, Table 1). Linear regression models with SWA and speed as predictors maximized the variance accounted for (Adj. R2) and improved the accuracy of the estimates (SEE). However, lowest AICc values were observed with the linear regression model that included only the SWA to predict IC EE (Table 2, Fig. 3).

85% VO2maxpred speed Running Outcomes. Device differences were not as apparent during the intervals when participants ran at 85% (0.05 kcal·min-1, 95% CI[-1.19, 1.08]. Absolute agreement at 85% VO2maxpred running speed was good (Fig. 1, Table 1). Like lower speed running, models with SWA and speed as predictors maximized the variance accounted for and minimized SEE. Also, like lower speed running, the lowest AICc values were observed with the linear regression model that included on the SWA alone to predict IC EE (Table 2, Fig. 3).

328-       Lack of consideration of slow component behavior temper the strength of these conclusions.  Please reword in line with consideration of the literature in this regard.

We did not calculate the VO2 slow component as energy systems shift because we did not perform a max test. We are trying to use standardized equations to approximate metabolic energy expenditure and then compare it to the BodyMedia’s energy expenditure approximations we got from the proprietary software.

We do acknowledge a limitation of not doing a max test and not assessing the VO2 slow component. We have changed the limitations section to address this for readers. It now states:

“Further, because a VO2 max test was not performed, the results are limited to the relative versus absolute effects of exercise intensity on EE estimates from the SWA. Without a max test, the Critical Speed[33] and role of the slow component[34] could not be accurately assessed, which may have led to an underestimation of EE with IC.”

Reviewer 2 Report

Comments and Suggestions for Authors

Thank you for giving me an opportunity to review the manuscript titled: “Accuracy of the SenseWear Armband During Short Bouts of Exercise”. Please find below my comments and suggestions for improvement.

Title: Shouldn’t the TM sign be above SenseWear instead of Armband?

Line 21: Please correct the formatting of the measurement units. Also, please rephrase “not significantly so”.

Line 25: How was the good agreement established when you indicated “but do not directly reflect”? Please reformat this sentence.

Line 61: Switch “more mobile” to “portable”. The same remark applies to the “wearer”. Please modify this word.

Line 84: Instead of using “we” terminology, I would suggest authors switch to using “the finding of the present study…” or similar.

Line 96: Just start with “20 participants volunteered to participate in the present study” or similar. Also, you need to include the age, weight, and height of the participants.

Line 98: What do you consider physically active? How was this determined?

Line 106: Did participants have a familiarization session?

Line 133: I would suggest authors break the methods section into separate sub-sections. This will help readers with their understanding of the overall article. For example, 2.1. participants, 2.2. procedures, 2.3. dependent variables, etc.

Line 169: Why did you not use ICC for the agreement between the measurements?

Line 181: You have not mentioned ICC in the statistical analysis section of the manuscript.

Line 214: Please correct the table to adhere to Sports guidelines. The same comment applies to the rest of the figures and tables included in the manuscript.

Line 231: As previously indicated, please avoid using “we” terminology. This comment applies to the rest of the manuscript.

Line 305: I would suggest authors consider mentioning the anatomical location at which sensor is placed. Please consider discussing/including the below-listed study or similar.

Cabarkapa, D. V., Cabarkapa, D., Philipp, N. M., & Fry, A. C. (2023). Impact of the Anatomical Accelerometer Placement on Vertical Jump Performance Characteristics. Sports, 11(4), 92.

Line 332: Delete typo at the end of the sentence.

Comments on the Quality of English Language

N/A

Author Response

Reviewer 1

Thank you for giving me an opportunity to review the manuscript titled: “Accuracy of the SenseWear Armband During Short Bouts of Exercise”. Please find below my comments and suggestions for improvement.

Thank you for reviewing our manuscript. We believe it is stronger after responding to the requested edits.

Title: Shouldn’t the TM sign be above SenseWear instead of Armband?

The TM symbol was removed from the title and in other locations throughout the manuscript to conform with other publications that studied the SenseWear Armband recently. For example, an article in this journal in 2017 (Ong et al.) used the TM symbol, but in 2019 (O’Brien et al.) did not use the symbol. We assume the trademark was not renewed because the business was acquired by Jawbone in 2013 and the product did not continue. However, we are happy to defer to the editor’s recommendation.

Line 21: Please correct the formatting of the measurement units. Also, please rephrase “not significantly so”.

The phrase has been changed to read “… Armband, but not significantly at…” and the formatting the units has been changed in the entire manuscript so “Kcal” is now “kcal”.

Line 25: How was the good agreement established when you indicated “but do not directly reflect”? Please reformat this sentence.

Thank you for the comment. The level of agreement was established from the ICC value. We have modified the sentence to read:

...displayed a good level of agreement (ICC=0.76 to 0.84) with IC measures.”

Line 61: Switch “more mobile” to “portable”. The same remark applies to the “wearer”. Please modify this word.

“More mobile” has been switched to “portable” and “wearer” has been switched to “user”.

Line 84: Instead of using “we” terminology, I would suggest authors switch to using “the finding of the present study…” or similar.

First person terminology was removed in all instances throughout the manuscript.

Line 96: Just start with “20 participants volunteered to participate in the present study” or similar. Also, you need to include the age, weight, and height of the participants.

Thank you for the comment. We changed the passage to read “Twenty physically active runners, free of any known disorders (e.g., cardiovascular, neuromuscular) that affected their ability to exercise or were exacerbated by exercise, volunteered for this study.” but did not include patient demographics here because it is included in the results section.

Line 98: What do you consider physically active? How was this determined?

The runners were determined to be recreational active based on the Perceived Functional Ability (PFA) score of ≥16/26 (i.e., 11-12 minutes per mile running pace), a Physical Activity Rating (PAR) of ≥ 6/10 (i.e., runs ≥ 5 miles per week), and a treadmill comfort score of ≥ 7/10 (i.e., completely comfortable). Highlighted verbiage in the 2.1 Participants section was updated to clarify, “To ensure safety with the study protocol, all volunteers were required to be recreationally active, defined as having a Perceived Functional Ability (PFA) score of ≥16/26 (i.e., 11-12 minutes per mile running pace), a Physical Activity Rating (PAR) of ≥ 6/10 (i.e., runs ≥ 5 miles per week), and a treadmill comfort score of ≥ 7/10 (i.e., completely comfortable).

Line 106: Did participants have a familiarization session?

The participants did not have a familiarization session. The participants had a history of running each week and were completely comfortable running on a treadmill as measured by a treadmill comfort score.

Line 133: I would suggest authors break the methods section into separate sub-sections. This will help readers with their understanding of the overall article. For example, 2.1. participants, 2.2. procedures, 2.3. dependent variables, etc.

Thank you for the suggestion. We have added sub-section titles in the methods section (Participants, Procedures, Dependent Variables, Statistical Analysis).

Line 169: Why did you not use ICC for the agreement between the measurements?

We used ICC to measure absolute agreement between the technologies (SenseWear Armband versus indirect calorimetry). The following statement was included in the original submission but may have been overlooked, “For each condition, intraclass correlations (ICC (3,1)) were computed to examine level of absolute agreement between the technologies.” We added the following highlighted text to draw attention to this approach, “Additionally, for each condition…” Please advise if more clarification is needed.

Line 181: You have not mentioned ICC in the statistical analysis section of the manuscript.

Please see our response to the related comment for previous Line 169. ICC was mentioned in the methods section of the last submission but may have been overlooked. We have added transitions between approaches in the methods to better draw attention to the ICC approach. Additionally, we hope the new sub headers in the methods section also improve readability.  

Line 214: Please correct the table to adhere to Sports guidelines. The same comment applies to the rest of the figures and tables included in the manuscript.

Thank you for pointing this out. The tables and figures have been modified to meet Sports guidelines.

Line 231: As previously indicated, please avoid using “we” terminology. This comment applies to the rest of the manuscript.

We have removed first person language throughout the manuscript. 

Line 305: I would suggest authors consider mentioning the anatomical location at which sensor is placed. Please consider discussing/including the below-listed study or similar.

Cabarkapa, D. V., Cabarkapa, D., Philipp, N. M., & Fry, A. C. (2023). Impact of the Anatomical Accelerometer Placement on Vertical Jump Performance Characteristics. Sports, 11(4), 92.

Thank you for the recommendation. We have included that reference and added to our limitations by stating:

“Lastly, because accelerometer location affects measurement of movement characteristics [35], EE estimation may have been affected by the sensor location (left triceps) prescribed by the user manual.”

Line 332: Delete typo at the end of the sentence.

Thank you for noticing our error. “6. Patents” was removed because that section is not applicable to this manuscript.

Reviewer 3 Report

Comments and Suggestions for Authors

The study demonstrated that SWA overestimated energy expenditure at lower intensities and, conversely, underestimated energy expenditure at higher movement intensities. Unfortunately, the VO2max value and the respective maximum aerobic running speed were not directly determined in the study, and only estimates of both parameters were used, which reduces the informative value of the results of the study. It can also be assumed that at higher velocities (e.g. 95 %), not only aerobic but also anaerobic component (post-exercise VO2) of energy expenditure should also be monitored when using indirect calorimetry and assessing energy expenditure.

Author Response

Reviewer 3

The study demonstrated that SWA overestimated energy expenditure at lower intensities and, conversely, underestimated energy expenditure at higher movement intensities. Unfortunately, the VO2max value and the respective maximum aerobic running speed were not directly determined in the study, and only estimates of both parameters were used, which reduces the informative value of the results of the study. It can also be assumed that at higher velocities (e.g. 95 %), not only aerobic but also anaerobic component (post-exercise VO2) of energy expenditure should also be monitored when using indirect calorimetry and assessing energy expenditure.

Thank you for the review and comment on our manuscript. We did not perform a max test because the purpose of the study was to compare the two measurement techniques at the same time, and we used standardized procedures to determine the predicted VO2max. The hope of using the SenseWear Armband is that it could accurately estimate energy expenditure at any intensity, whether it is a predicted 75% VO2max pace or an actual 75% VO2max pace from a max test. From there, predictions of which energy systems could be made but was not the goal of this study. We will consider performing a max test for future studies and measuring the post-exercise VO2 as well.

We have included the following limitations:

“Further, because a VO2 max test was not performed, the results are limited to the relative versus absolute effects of exercise intensity on EE estimates from the SWA. Without a max test, the Critical Speed [33] and role of the slow component [34] could not be accurately assessed, which may have led to an underestimation of EE with IC.”

Reviewer 4 Report

Comments and Suggestions for Authors

Dear Editor,

Thank you for the opportunity to review the manuscript “Accuracy of the SenseWear ArmbandTM During Short Bouts of Exercise” by Wedge et al. This manuscript examines the accuracy of the SenseWear Armband for estimating energy expenditure during short bouts of walking and running at different intensities compared to indirect calorimetry.

Overall, this is a well-written manuscript on an important topic in exercise science and wearable technology. The methods are sound and the results are clearly presented. I have some general and specific comments for the authors to address that I believe will strengthen the quality and clarity of the manuscript.

General comments

  • The introduction provides good background and rationale for the study aims. More information is needed on the SenseWear Armband technology and how it estimates energy expenditure.
  • The sample size is relatively small at 20 participants. Discussion of this limitation would be helpful.
  • More detail is needed in the results section on the actual energy expenditure values and differences between measurement methods. Inclusion of a table may help summarize the key outcome data.
  • The discussion and conclusion sections which overstate the accuracy and usefulness of the SWA. While the SWA performed reasonably well at higher intensities, it significantly overestimated EE for walking. The authors need to better acknowledge this limitation throughout the manuscript.

Specific comments

Abstract

  • Define acronym EE first before using abbreviation.
  • The statement "the SenseWear Armband may enable energy expenditure estimation in free-living settings..." is too strong based on a lab study using prescribed locomotor activities.

Introduction

  • Provide more technical detail on what physiological signals the SenseWear Armband measures and how the proprietary algorithm uses that data to estimate EE.
  • Provide some example values for errors in EE estimation to characterize level of accuracy needed.
  • Specify earlier reports that have questioned SWA's accuracy during intermittent bouts (e.g. reference #11). Please, authors need to clearly identify the gaps addressed by the study - specifically the accuracy of the SWA for short duration high-intensity activity up to 95% VO2max.

Methods

  • Clarify how time synchronization was performed between the SenseWear Armband and indirect calorimetry systems.
  • Confirm that the university's IRB approved all study procedures.
  • Were participants given adequate familiarization to the testing protocols prior to data collection?

Results

  • Consider reorganizing parts of the results section for better flow rather than jumping between statistical tests - first summarize group/condition outcomes, then regression results.
  • Figure 1: Display mean EE values for each condition on the graph to facilitate comparisons
  • Add a table detailing i) the mean EE values per method and ii) mean differences between methods with efforts sizes and significance values for each condition.
  • Add measures of variance (e.g. SD) alongside the mean values.

Discussion

  • The statement on SWA's accuracy for between-subject comparisons is speculative here and needs context from cited studies.
  • The statement about "not justifying extra sensors" seems premature without considering factors like cost, feasibility, usage burden of additional monitoring.
  • Provide more details on the discontinued SWA model tested and the status of the technology going forward.
  • Discuss study limitations related to sample size and population

Conclusions

  • The conclusions regarding accuracy for repeated testing and utility for monitoring training workload overreach what the study itself has shown. The study design does not directly assess test-retest reliability.

I hope these comments are helpful for the authors. With some minor revisions to address the clarity and presentation of methods and results, this manuscript can provide a valuable contribution on this topic. Please feel free to contact me with any questions.

I look forward to reviewing the revised manuscript.

Sincerely,

Author Response

Thank you for the opportunity to review the manuscript “Accuracy of the SenseWear ArmbandTM During Short Bouts of Exercise” by Wedge et al. This manuscript examines the accuracy of the SenseWear Armband for estimating energy expenditure during short bouts of walking and running at different intensities compared to indirect calorimetry.

Overall, this is a well-written manuscript on an important topic in exercise science and wearable technology. The methods are sound and the results are clearly presented. I have some general and specific comments for the authors to address that I believe will strengthen the quality and clarity of the manuscript.

Thank you for reviewing our manuscript. We believe it is stronger after responding to the requested edits.

General comments

The introduction provides good background and rationale for the study aims. More information is needed on the SenseWear Armband technology and how it estimates energy expenditure.

The sample size is relatively small at 20 participants. Discussion of this limitation would be helpful.

More detail is needed in the results section on the actual energy expenditure values and differences between measurement methods. Inclusion of a table may help summarize the key outcome data.

The discussion and conclusion sections which overstate the accuracy and usefulness of the SWA. While the SWA performed reasonably well at higher intensities, it significantly overestimated EE for walking. The authors need to better acknowledge this limitation throughout the manuscript.

Thank you for these general comments. We have addressed these comments through the specific comments below and feel the manuscript is much improved. Thank you for your suggestions.

Specific comments

Abstract

Define acronym EE first before using abbreviation.

Thank you for pointing this out. EE is now defined in the first sentence of the abstract.

The statement "the SenseWear Armband may enable energy expenditure estimation in free-living settings..." is too strong based on a lab study using prescribed locomotor activities.

We have modified the statement to read:

The SenseWear Armband may enable EE estimation of locomotion outside the laboratory, including…

Introduction

Provide more technical detail on what physiological signals the SenseWear Armband measures and how the proprietary algorithm uses that data to estimate EE.

Thank you for this comment. We provided more detail as best we could, and it now reads:

“The SenseWear Armband (SWA), developed by BodyMedia, is a relatively unobtrusive sensor worn on the upper arm during physical activity and collects information about near-body ambient temperature, skin temperature through a proprietary heat flux sensor, triaxial acceleration, and galvanic skin response which varies due to physical and emotional stimuli. Output from each of these signals is available with the SWA Professional software, but how the software uses these measures along with demographic characteristics (i.e., age, weight, height, sex, handedness, smoker or non-smoker) in a proprietary algorithm to estimate EE in free living conditions is unknown. The SWA Professional software outputs EE estimations each minute using this unshared proprietary algorithm.”

Provide some example values for errors in EE estimation to characterize level of accuracy needed.

Thank you for this comment. We have provided an example of errors in EE estimation from the literature. It now states: “Results from Gastin et al. [11] showed a 36.7% difference for walking, 14.9% difference for running (12 km·h-1), and 36% difference for circuit training when comparing the SWA to IC.”

Specify earlier reports that have questioned SWA's accuracy during intermittent bouts (e.g. reference #11). Please, authors need to clearly identify the gaps addressed by the study - specifically the accuracy of the SWA for short duration high-intensity activity up to 95% VO2max.

Thank you for these comments. We added details to clarify this gap and provide specific examples. The introduction now includes:

Also, results from van Hoye et al. [12] suggest accuracy of the SWA compared to IC worsens with increasing speeds of an incremental exercise test; error increased from 2-5% at 1.5 m·s-1 up to 39-46% at near max 4.0 m·s-1, with the largest errors (>19%) seen at running speeds ≥ 3.0 m·s-1 [12]. These results are consistent with other studies that report the SWA underestimates EE during high intensity physical activity [12,15–17]. One reason may be that its proprietary algorithm does not account for heart rate (HR), which has been shown to be an important predictor of EE using sensor data [6]. Because EE is directly related to HR and speed with running, incorporating additional sensors capable of measuring HR and speed could improve the accuracy of EE estimations [18,19] and represents a gap to be addressed.”

Methods

Clarify how time synchronization was performed between the SenseWear Armband and indirect calorimetry systems.

We synchronized based on the seconds digit for GMT-5 time stamps from both systems. The passage now states:

“During the testing protocol, SWA and IC data were collected continuously with timestamps based on GMT-5 to allow for time synchronization between devices.”

Confirm that the university's IRB approved all study procedures.

We have added this to the manuscript and it now states:

“The study was conducted in accordance with the Declaration of Helsinki, all procedures performed during the study were approved by the local University Institutional Review Board (UMCIRB 15-002097) and all participants provided written informed consent after being told the risks and benefits of the study.”

Were participants given adequate familiarization to the testing protocols prior to data collection?

The participants were given time to become familiar with the testing environment by warming up for 5 minutes with a gradually increasing speed based on their comfort. We also used a treadmill comfort score of ≥ 7/10, which indicates being completely comfortable, and the participants had a history of running each week.

Results

Consider reorganizing parts of the results section for better flow rather than jumping between statistical tests - first summarize group/condition outcomes, then regression results.

Thank you for this feedback. To improve readability, sub headers were created to present data by conditions, and with the addition of a table, we think the results are clearer.

Figure 1: Display mean EE values for each condition on the graph to facilitate comparisons

Figure 1 is a violin plot which has the median value in the open white circle in the middle of each plot and shows the spread of the data by displaying each participant’s data for that condition and measurement (IC vs. SWA). We have made median circle and the interquartile range box larger to improve visibility. We have made the median circle and the interquartile range box larger to improve visibility. To facilitate comparisons of EE for each condition, we have added a table of mean values, differences, effect sizes, significance values, and ICC values as suggested below.

Add a table detailing i) the mean EE values per method and ii) mean differences between methods with efforts sizes and significance values for each condition.

Thank you for the suggestion. We have created a new table, and it is now Table 1.

Add measures of variance (e.g. SD) alongside the mean values.

Thank you for the suggestion. We have included the 95% confidence limits of the mean differences in the narrative and included the SD of the mean difference in the new Table 1.

Discussion

The statement on SWA's accuracy for between-subject comparisons is speculative here and needs context from cited studies.

We did not examine or report between subjects’ comparisons. To make this evident, we added the highlighted text:

Somewhat consistent with our first hypothesis, the SWA estimated EE for different exercise intensities with good accuracy across subjects, except walking.”

If this does not address the reviewer’s concerns, clarification of this comment from the reviewer would be greatly appreciated.

The statement about "not justifying extra sensors" seems premature without considering factors like cost, feasibility, usage burden of additional monitoring.

Thank you for pointing this out. We did have a section addressing this topic in the fourth paragraph of the discussion section:

If the goal is to accurately estimate EE, such as with athletes or serious exercisers, extra sensors that capture data related to EE, like HR and speed [18,19], may be necessary. Ideally extra sensors would be integrated into one sensor to increase acceptance across users.”

However, we have modified the last sentence in the first paragraph of the discussion to ensure this message is clear:

“Inclusion of speed and HR in future devices depends on cost, unification into one device, and burden to the user.”

Provide more details on the discontinued SWA model tested and the status of the technology going forward.

We found details online about the acquisition of BodyMedia by Jawbone and cannot find any related products released by Jawbone. We have added this statement:

“BodyMedia was purchased by Jawbone in 2013 and the product was not pursued further, therefore, development of a new device incorporating multiple physiological signals may be needed.”

Discuss study limitations related to sample size and population

We have restructured the limitation section of the discussion and added this statement:

“We determined that 20 was a large enough sample to detect device differences based on an a priori power analysis and preliminary lab data but acknowledge the limitations of this sample size for reliability analysis and exploratory prediction equations. Studying a homogenous population limits generalizability.”

Additionally, we have also moved and added text related to homogeneity of our population and potential sex differences to this section.

Lastly, while not added to the manuscript, we retrospectively compared of our multiple correlation coefficients to Knofczynski & Mundfrom (2008) Table 2 sample size recommendations based on number of predictors. Our squared multiple correlation coefficients of 0.4-0.7 correspond to a sample size recommendation of 7-22 participants.

Knofczynski, G. T., & Mundfrom, D. (2008). Sample sizes when using multiple linear regression for prediction. Educational and Psychological Measurement, 68(3), 431–442. https://doi.org/10.1177/0013164407310131

Conclusions

The conclusions regarding accuracy for repeated testing and utility for monitoring training workload overreach what the study itself has shown. The study design does not directly assess test-retest reliability.

Thank you for this comment. We have softened the language and it now states:

“Except for walking, small absolute differences relative to IC were observed. The SWA has potential to be a useful tool for monitoring workload and fitness progress outside the lab in sports and activities that involve short intervals, particularly within session, but further testing should be performed to determine repeated testing capabilities.”

I hope these comments are helpful for the authors. With some minor revisions to address the clarity and presentation of methods and results, this manuscript can provide a valuable contribution on this topic. Please feel free to contact me with any questions.

Thank you for your feedback. Your time and effort in reviewing this manuscript is greatly appreciated.

I look forward to reviewing the revised manuscript.

Round 2

Reviewer 1 Report

Comments and Suggestions for Authors

Whereas some rebuttal statements may be considered evasive, the authors have revised the text sufficiently to improve clarity and science presentation.

Reviewer 4 Report

Comments and Suggestions for Authors Dear authors,   Thank you for submitting the revised manuscript "Accuracy of the SenseWear ArmbandTM During Short Bouts of Exercise". I have reviewed the authors' responses to the comments and the revised manuscript.

The authors have thoroughly addressed all of the general and specific comments raised during the initial review. They have provided additional details on the SenseWear Armband technology, acknowledged the study's limitations, reorganized the results section for improved clarity, included a new table summarizing the key outcome data, and tempered some of the conclusions to better align with the study findings.

The authors' responses are satisfactory, and the revisions have significantly improved the quality and clarity of the manuscript. The comments have been adequately addressed, and the manuscript is now suitable for publication in its present form.

I commend the authors for their diligent efforts in responding to the reviewers' comments and making the necessary revisions. This manuscript provides valuable insights into the accuracy of the SenseWear Armband for estimating energy expenditure during short bouts of exercise, which will be of interest to researchers and practitioners in the field of exercise science and wearable technology.

I recommend accepting the revised manuscript for publication.

Kindly regards